



# How intense daily precipitation depends on temperature and the occurrence of specific weather systems – an investigation with ERA5 reanalyses in the extratropical Northern Hemisphere

Philipp Zschenderlein[1] and Heini Wernli[1]

[1]Institute for Atmospheric and Climate Science, ETH Zurich, Universitätstr. 16, 8092 Zurich, Switzerland

**Correspondence:** philipp.zschenderlein@env.ethz.ch

**Abstract.** Precipitation and surface temperature are two of the most important variables that describe our weather and climate. Several previous studies investigated aspects of their relationship, for instance the climatological dependence of daily precipitation on daily mean temperature, $P(T)$. However, the role of specific weather systems in shaping this relationship has not been analysed yet. This study therefore identifies the weather systems (WS) that are associated with intense precipitation days as a function of $T$, focusing on the question how this relationship, symbolically expressed as $P(T,\text{WS})$, varies regionally across the Northern Hemisphere and between seasons. To this end, we first quantify, if intense precipitation occurs on climatologically warmer or on colder days, respectively. In winter, over most continental and ocean regions, intense precipitation falls on warmer days apart from the Mediterranean area and regions in the lee of the Rocky Mountains, where intense precipitation is favoured on colder days. In summer, only at high latitudes intense precipitation is favoured on warmer days, whereas continental areas experience intense precipitation on colder days. For selected regions in Europe and North America, we then identify the weather systems that occur preferentially on days with intense precipitation (referred to as wet days). In winter, cyclones are slightly dominant on colder wet days, whereas warm conveyor belts and atmospheric rivers occur preferentially on warmer wet days. In summer, the overall influence of atmospheric rivers increases and the occurrence of weather systems depend less on wet day temperature. Wet days in the lee of the Rocky Mountains are influenced by most likely convective systems in anticyclones. Finally, we investigate $P(T,\text{WS})$ during the wettest and driest season in Central Europe and the Central US. In qualitative agreement with the results from the first part of this study, the wettest winter is warmer than normal in Central Europe but colder in the Central US, and the wettest summer is colder in both regions. The opposite holds for the driest winter and summer, respectively. During these anomalous seasons, both the frequency and the precipitation efficiency of weather systems changes in Central Europe, while the wettest and driest seasons in Central US mainly arise from a modified precipitation efficiency. Our results show that the precipitation-temperature-weather system relationship strongly depends on the region, and that (extreme) seasonal precipitation is influenced by the frequency and precipitation efficiency of the different weather systems. This regional variability is reflected in the relative importance of weather system frequency and efficiency anomalies for the formation of anomalously wet and dry seasons.



## 1 Introduction

Anomalously wet and dry seasons have a large impact on society, economy, and the environment (IPCC, 2012). The wet winter 2013/2014 in the United Kingdom led to severe flooding (Kendon and McCarthy, 2015), while the dry summer 2010 in Russia was associated with a long-lasting heat wave, widespread wildfires, and increased mortality rates (Dole et al., 2011; Shaposhnikov et al., 2014). Both seasons were not only accompanied by anomalously high and low precipitation, but also by warmer temperatures than normal, which points to the co-variability of surface temperature, $T$, and precipitation, $P$. This relationship

has received considerable attention in the scientific literature. Madden and Williams (1978) analysed the correlation between seasonal mean 2-m temperature and precipitation for the United States and Europe and found that cool summers are typically wet in both regions and warm winters are typically wet over the eastern United States and large parts of Europe. In a global analysis, Trenberth and Shea (2005) found negative correlations between monthly mean temperature and precipitation over most land areas in both seasons and positive correlations for high latitudes in the cold season. Isaac and Stuart (1992) investigated

the co-variability with daily data from Canadian weather stations and showed that wet days are typically colder than average in summer, whereas in winter, wet days are warmer at the east and west coast, while areas in the lee of the Rocky Mountains experience precipitation preferentially on the colder days in all seasons. Later, many studies analysed the relationship between surface temperature and extreme precipitation with observational datasets (e.g., Lenderink and Van Meijgaard, 2008; Berg et al., 2009; Hardwick Jones et al., 2010; Utsumi et al., 2011; Panthou et al., 2014; Drobinski et al., 2016; Gao et al., 2018).

They found that the intensity of extreme precipitation can either increase or decrease with temperature, stay constant at higher temperatures, or it can even increase initially and then decrease again at higher temperatures [see overview in Panthou et al. (2014)]. The observed relationship between temperature and extreme precipitation is shaped by the dynamics of the rainfall events, or, more explicitly, by the duration of the events, as shown for Canadian stations by Panthou et al. (2014). For example, rainfall events lasting longer than 12 h show a less strong increase of rainfall intensity with temperature than events lasting

less than 1 h. Also the precipitation type and season can influence the co-variability of temperature and precipitation extremes. Drobinski et al. (2016) observed, for the French Mediterranean region, a gradual increase of frontal precipitation intensity with temperature in winter and a decline of convective precipitation intensity with temperature in summer.

Although the relationship between temperature and precipitation [hereafter denoted as $P(T)$] has been studied extensively on daily and seasonal time scales, it is less clear how this relationship is influenced by the occurrence of different weather

systems. The unusually wet and mild winter 2013/2014 in the United Kingdom was accompanied by an enhanced frequency of extratropical cyclones (Davies, 2015; Kendon and McCarthy, 2015) – a weather system that is often associated with (extreme) surface precipitation (Field and Wood, 2007; Pfahl and Wernli, 2012). The fact that, in the UK, the extremely wet winter was also anomalously warm goes along with the concepts that, in general, warmer air can hold more moisture and, more specifically, that warm air advection at the warm front of the cyclone is typically also moist (Catto and Pfahl, 2013). However, precipitation

at the cold front of the cyclone can also lead to the combination of cold and wet conditions, which prevailed during the summer 2012 in northern Europe (Yiou and Cattiaux, 2013). Whether an anomalously wet season is warmer or colder than normal therefore appears to be related to the frequency of occurrence of particular weather systems (e.g., precipitation in the warm




sector of a cyclone vs. in the cold air behind cold fronts) and, more basically, whether precipitation falls predominantly on the colder or on the warmer days of a season. This study investigates the extended relationship of precipitation on temperature and on the occurrence of weather systems (WS), symbolically expressed as $P(T, \text{WS})$, using ERA5 reanalysis data and with a focus on the extratropical northern hemisphere in winter and summer. Our hypothesis is that this relationship may vary strongly between regions and seasons. An important choice that is required to conduct such an analysis, is the selection of a meaningful set of weather systems, which can be objectively identified. Extratropical cyclones are an obvious choice, and we decided to also consider warm conveyor belts, atmospheric rivers, and convection in anticyclones, as outlined in the next paragraph.

Apart from cyclones, also other weather systems are known to be related to precipitation, e.g. warm conveyor belts. They are coherent, poleward ascending airstreams at the cold or warm front of extratropical cyclones (e.g., Browning et al., 1973; Wernli, 1997). During the ascent from the planetary boundary layer to the upper troposphere, the air cools and water vapour condenses, and warm conveyor belts therefore contribute essentially to the cloud structure and precipitation of cyclones (e.g., Browning, 1990). Warm conveyor belts are also often associated with extreme precipitation (Pfahl et al., 2014). Climatologically, warm conveyor belts occur less frequently than cyclones, i.e., not every extratropical cyclone has a warm conveyor belt, and they occur preferentially in the entrance regions of the storm tracks (Madonna et al., 2014). Atmospheric rivers are also often associated with extreme precipitation, especially at the west coast of the United States (US) (Roberge et al., 2009; Rutz et al., 2014). They are defined as long, narrow, and transient filaments of both enhanced vertically integrated water vapour (IWV) and vertically integrated horizontal water vapour transport (IVT), and they are often associated with a low-level jet ahead of the cold front of an extratropical cyclone (Ralph et al., 2018). Atmospheric rivers, identified with fixed thresholds of IWV and IVT, occur most frequently over the North Pacific and North Atlantic, with a generally higher frequency in summer (Sodemann et al., 2020). It should be noted that these weather systems are not mutually exclusive and can therefore co-occur. As an example, a warm conveyor belt ascends in the vicinity of an extratropical cyclone (Browning, 1990; Madonna et al., 2014) and atmospheric rivers can be located in the warm sector of a cyclone ahead of the cold front (e.g. Fig. 16 in Ralph et al., 2017) and overlap with the inflow and ascent regions of a warm conveyor belt (Sodemann et al., 2020).

Apart from these large-scale weather systems, also mesoscale processes, such as deep convection or orographic ascent can be associated with intense precipitation. Often, large-scale weather systems provide favourable conditions for the triggering of intense precipitation by the regional-scale circulation. The combined effects of large-scale weather systems and mesoscale processes are, for example, observed near the Rocky Mountains, where a suitably positioned surface anticyclone can induce a warm and moist low-level jet directed from the southwest towards the Rocky Mountains, and the combination of orographic lifting and convection can then lead to extreme precipitation. In addition, the weak mid- to upper-level flow favours slow-moving convective systems and can further increase rainfall accumulation (Pontrelli et al., 1999; Lin et al., 2001). Therefore, we also identify surface high-pressure systems and regions with elevated values of CAPE in this study.

The main objective of our study is to introduce the weather system perspective into the $P(T)$ relationship and to analyse, which weather systems occur preferentially in certain temperature ranges on days with intense precipitation in winter and summer. This knowledge is then used to analyse anomalously wet and dry seasons. More specifically, we address the following research questions in the extratropical Northern Hemisphere:



1. Does intense precipitation typically occur on the colder or on the warmer days of a season?

2. Which weather systems occur preferentially on days with intense precipitation, and do some weather systems typically occur on colder or warmer wet days?

3. In anomalously wet and dry seasons, how do the frequency of occurrence of weather systems and their precipitation efficiency differ from climatology?

For all these questions, we are curious to identify regional and seasonal differences.

Section 2 provides an overview of the data and methods employed in this study. Section 3 investigates the $P(T)$ relationship in the extratropical Northern Hemisphere (question 1). The weather systems that occur on wet days as a function of temperature are presented in Section 4 (question 2). Section 5 then focuses on the $P(T,WS)$ relationship in anomalous seasons (question 3). In Section 6, a summary of the main findings and avenues for further research are presented.

## 2 Data and Methods

This section first introduces the methodology to quantify whether intense precipitation occurs mainly on the warmer or on the colder days. Afterwards, we describe the identification of the weather systems considered in this study and the attribution of these systems to days with intense precipitation. The section closes with a description of how anomalous seasons are identified. If not noted otherwise, all analyses are based on the ERA5 reanalysis of the European Centre for Medium-Range Weather Forecasts (Hersbach et al., 2020) interpolated to a $0.5° \times 0.5°$ longitude-latitude grid for the period 01 December 1980 to 30 November 2019.

### 2.1 Intense precipitation as a function of temperature

In order to identify whether intense precipitation occurs on warmer or on colder days of the local climatological temperature distribution, we compare the probability distributions of temperature on all days and temperature on days with intense precipitation. First, we calculate at every grid point the daily mean 2-m temperature and daily accumulated total precipitation, which is the sum of convective and large-scale precipitation. Secondly, we define days with intense precipitation as days when the daily accumulated precipitation exceeds the 95th percentile, which is calculated separately for every grid point and season. The 95th percentile leads to 179 intense precipitation days per grid point in winter and summer, respectively, in the considered 39 years, i.e. on average 4-5 per individual season. Higher percentiles would result in a lower number of events and therefore a decreased statistical robustness. Thirdly, at every grid point and in every season, we calculate the fraction of intense precipitation days whose mean temperature is below the local median of daily mean temperatures, abbreviated with WBM, (Fig. 1a), following the approach of Isaac and Stuart (1992). If the WBM value is below 50% at a grid point, then intense precipitation occurs predominantly on warmer days. The lower (higher) this percentage, the more robust is the conclusion that intense precipitation occurs on warmer (colder) days. In the schematic example shown in Fig. 1a considerably less than 50% of intense precipitation





days have a mean temperature below the local median, which implies that intense precipitation occurs predominantly on the warmer days. In the following, we will refer to days with intense precipitation as wet days for the sake of brevity.

## 2.2 Weather systems

All weather systems are identified as six-hourly two-dimensional binary fields with the value of 1 at grid points that belong to the weather system and the value of 0 at grid points that do not belong to it. The identification of cyclones, anticyclones and warm conveyor belts is based on the methods described in Sprenger et al. (2017), and for atmospheric rivers the method of Sodemann et al. (2020) is used. Note that the climatologies of these weather systems were originally computed with ERA-Interim data and have been recalculated with ERA5 for the purpose of this study. The next paragraph briefly describes the main elements of these identification algorithms; further details can be found in the specific references.

Surface cyclones and anticyclones are defined as regions delimited by the outermost closed sea level pressure (SLP) contour. The contour must enclose at least one minimum for cyclones, or maximum for anticyclones (Wernli and Schwierz, 2006). Warm conveyor belts are trajectories ascending at least 600 hPa in 2 days from the planetary boundary layer to the upper troposphere near an extratropical cyclone (Madonna et al., 2014). We use 2D binary fields that represent the WCB ascent, which corresponds to the horizontal envelope of the positions of the WCB-trajectories during their ascent between 800 and 400 hPa. This ascent phase is the part of a WCB with the strongest precipitation (Madonna et al., 2014). Atmospheric rivers are identified at grid points where (i) IWV $> 20\,\mathrm{kg\,m^{-2}}$ (equivalent to 20 mm), and (ii) IVT $> 250\,\mathrm{kg\,m^{-1}\,s^{-1}}$. The 2D objects of atmospheric rivers must have a horizontal extension of at least 2000 km and must be poleward of 20 °N. This definition was used in Sodemann et al. (2020) and agrees with the definition of Ralph et al. (2018).

At every grid point, we attribute what we regard as the dominant weather system to each wet day. Allowing for a small spatial displacement between the weather systems and the precipitation, a $2° \times 2°$ box centred around the grid point with intense precipitation is used to quantify the occurring weather systems. The system that covers the largest area of this $2° \times 2°$ box is considered as the dominant weather system. As an example, in Fig. 1b, the grey weather system is identified as the dominant one for the wet day at the red grid point. In order to increase the statistical robustness, this attribution analysis is not only performed for the wet day grid point, but for all 25 grid points (black points in Fig. 1b) in the $2° \times 2°$ box (Fig. 1c). As already mentioned in the Introduction, the identified weather systems can co-occur and their relative contribution to the daily accumulated precipitation cannot be quantified. Hence, our identification of a dominating weather system based on spatial overlap is a strong simplification, and it does not mean that this weather system exclusively led to the intense precipitation.

The weather system frequencies are analysed as a function of the wet-day temperature (Fig. 1d) for the $2° \times 2°$ box. We are particularly interested whether the $P(T)$ relationship differs for wet days attributed to different weather systems. Figure 1d shows such an example, where the grey weather system is dominating on colder wet days, while the blue weather system is more frequent on warmer wet days. In such a situation, we say that weather systems are relevant for the $P(T)$ relationship, and we introduce a measure called RWS (abbreviation for **R**elevance of **W**eather **S**ystems), which is calculated as follows. First, we compute a histogram like in Fig. 1d, and from this histogram, we calculate for each $T$-bin the percentage of counts ($f_i$) for each weather system, such that $\sum f_i = 1$. We do this only for 90% of the wet days in order to avoid very small counts at





the tails of the $T$-distribution, i.e. we only consider wet days between the 5th and 95th percentile of the $T$-distribution. Then, we calculate, for each weather system individually, the average $f$-values across all $T$-bins. With these values, we compute the anomalies of $f$ in each $T$-bin. In a last step, we calculate the RWS value by summing up all absolute $f$-anomalies and

multiplying the resulting value by 100. As the RWS value depends on the number of $T$-bins, we divide the RWS value by the number of $T$-bins. If the distribution of $f$-values is exactly the same for all $T$-bins, then the anomalies are zero for each $T$-bin, and the RWS value, i.e. the sum of all $f$-anomalies, is zero. Hence, in such a situation, the weather systems would not influence the $P(T)$ relationship. However, if RWS reaches a large value, then $P(T)$ differs for different weather systems, and we regard weather systems as relevant for understanding $P(T)$.

## 165 2.3 Identification of wettest and driest seasons

At every grid point and separately for winter and summer, the wettest and driest season is identified based on the area-averaged seasonally accumulated precipitation. More specifically, we calculate the seasonally accumulated precipitation at every grid point and then average these values over the $2° \times 2°$ box centred at the grid point (black box in Fig. 1c). As we are not interested in very local structures, the box-averaging is used to obtain representative values on the scale of 200 km. The season with the

maximum seasonal mean value is defined as the wettest season, and the season with the minimum as the driest season. Note that we take into account all days of a season when calculating the seasonal mean and not only precipitation on days with intense precipitation, i.e. wet days. Once the wettest and driest seasons are determined, we can analyse whether these seasons are warmer or colder than normal (i.e. the climatological mean), and if, during these anomalous seasons, weather systems occur more or less frequently and/or produce more or less precipitation. Further methodological details are presented at the

beginning of Section 5.

## 3 The $P(T)$ relationship

We start our analysis with the question whether wet days are predominantly warm or cold (compared to the local climatology) and how this differs regionally and seasonally.

### 3.1 The winter season (December-February, DJF)

In DJF, in most regions of the extratropical Northern Hemisphere, intense precipitation falls predominately (>80%) on warmer days, especially at higher latitudes, e.g. in the Arctic, Siberia, northern parts of Canada and Greenland (Fig. 2a). These areas experience very cold temperatures in winter due to the (partial) absence of incoming solar radiation and the presence of a widespread snow cover. According to the Clausius-Clapeyron relationship, saturation specific humidity is therefore very low in the high latitude winter and consequently also the 95th percentile of daily precipitation is low (Fig. 2c). In these regions, a

larger amount of precipitation, i.e. the occurrence of a wet day, requires the advection of warmer air with comparatively high specific humidity, which occasionally occurs in the form of moist intrusions (e.g., Bintanja et al., 2020) often associated with extratropical cyclones (e.g., Binder et al., 2017). Hence, it makes sense that wet days in the Arctic are almost exclusively warm.





Over most areas of the North Pacific and the North Atlantic, intense precipitation also falls predominantly on warmer days during winter (Fig. 2a), however, the signal over the North Pacific is more homogeneous compared to the North Atlantic, where

the fraction of warm wet days decreases from west to east from about 80 to 50%. Hence, the distribution of temperature on wet days and all days is rather similar over the eastern North Atlantic and therefore intense rain occurs equally likely on warm and cold days. Interestingly, there appears to be a correlation between the climatological wind speed at 300 hPa and the wet day frequency on warm days. The maxima of the jet stream over the western North Atlantic and the generally elevated wind speeds over the North Pacific are co-located with areas where intense precipitation is favoured on warmer days. In addition,

areas with a high 95th percentile of daily precipitation over the oceans agree with areas where precipitation falls on warmer days (Fig. 2a,c). One plausible explanation for this might be the increased baroclinicity in strong jet stream regions, which (i) is an important prerequisite for cyclone development, and (ii) facilitates intense precipitation where southerly (warm) flow is forced to ascend along the comparatively steep isentropes, e.g. in the form of a WCB. As an example, the dark red stripe in Fig. 2a over the western North Atlantic, an area known for its high cyclogenesis frequency (Wernli and Schwierz, 2006), is

co-located with a high 95th percentile of daily accumulated precipitation. However, it should be noted that this relationship between upper-level winds and wet day frequency is discernible only in winter (Fig. 2a and Fig. S1, which shows the same analysis for the spring and autumn seasons).

In contrast, the Mediterranean, characterised by an increased cyclone frequency (Wernli and Schwierz, 2006), experiences intense precipitation on colder days (60-90%, Fig. 2a). The wet day threshold is comparatively high (Fig. 2c). The other region

with cold wet days is the Central US in and downstream of the Rocky Mountains, which was also described in Isaac and Stuart (1992). In contrast to the Mediterranean, the wet day threshold in this area is quite low (Fig. 2c) due to lee effects of the Rocky Mountains. In addition, climatological wind speeds at 300 hPa are low in the vicinity of the Rocky Mountains as well as over the Mediterranean, which implies that the absence of a strong upper-level jet and therefore, low baroclinicity provides a favourable condition for intense precipitation to occur on colder days.

## 3.2   The summer season (June-August, JJA)

During JJA, most of the wet days are cold in many regions of the extratropical northern hemisphere (Fig. 2b). The signal is particularly strong with fraction values exceeding 90% in regions that are close to very dry summer climates, e.g. southern Europe, regions near the Black or the Caspian Sea, or the western US, but also in regions with a wet summer climate, e.g. the eastern US (Fig. 2d). Overall, the contrast between land and ocean is stronger during summer than winter, presumably due

to the weaker jet stream (Fig. 2b), i.e. the reduced role of baroclinic instability for the formation of precipitation systems and the increased role of continental convection and monsoonal flows that are driven by land-sea contrasts. It is interesting that in many regions over land, in contrast to winter, intense precipitation falls on the colder and not on the warmer days. This implies that high temperatures and a potentially high local moisture availability alone is not sufficient for intense precipitation. A first reason for this is that horizontal moisture transport into the area is relevant for intense precipitation also in the warm

season (e.g., Panthou et al., 2014; Grams et al., 2014). In addition, weather systems that occur during warmer days in summer, e.g. blocks and stationary anticyclones, suppress the formation of precipitation due to subsidence (e.g., Zschenderlein et al.,





2019). Only the very high latitudes of the North Pacific and the North Atlantic (with fractions of 60-70%), as well as Greenland (>80%), are characterised by preferentially warm wet days also in summer.

The pattern over the Arctic appears to be strongly influenced by the climatological sea ice extent in both winter and summer
(Fig. 2a,b). In winter, the fraction of comparatively warm wet days is particularly high over sea ice, locally exceeding 95%, which is plausible because precipitation in this area depends on the transport of warm and moist air. In summer, the pattern reverses and the climatological sea ice extent marks the border between areas where precipitation preferentially occurs on warm days over the North Pacific and North Atlantic and areas with precipitation on (mostly) colder days over the sea-ice covered Arctic. One possible reason for this pattern is that warmer summer days in the Arctic are accompanied by high-pressure systems
that suppress precipitation and increase solar radiation and sea ice melting (Wernli and Papritz, 2018).

### 3.3 Selected regional analyses

In order to better understand how intense precipitation depends on temperature, we compare the temperature distributions from all days and from wet days for three selected regions in Europe and North America (marked with yellow crosses in Fig. 2a,b; see Tab. 1 for coordinates and abbreviations) in winter and summer, respectively. In all regions, the shape of the overall distribution
of daily mean temperatures resembles, as expected, a (skewed) normal distribution with varying means and variances (black line in Figs. 3 and 4). Temperatures are less variable over the Mediterranean Sea (MED) and western US (W-US) compared to all other regions (Figs. 3c,f and 4b,e) and W-US experiences the smallest temperature contrast between summer and winter (Fig. 4b,e). The shape of the temperature distributions in the eastern US (E-US) and central US (C-US) differ substantially between the two seasons, especially E-US has a broad distribution in winter and a narrow one in summer (Fig. 4a,c,d,f). The
differing shapes are not only discernible for the whole temperature distribution, but also for the temperature distribution on wet days only, in particular for E-US and C-US, where the variance is higher in winter compared to summer. In general, the $T$-distribution of all days and of wet days only differ substantially in regions where the fraction values in Fig. 2a,b are either very high or very low, i.e. where wet days are predominantly cold or warm. As an example, more than 80% of wet days in winter in C-EU are warm (Figs. 2a and 3a), and therefore, the $T$-distributions of all days and wet days differ substantially in
Fig. 3a. In agreement with Fig. 2, the deviation of the two $T$-distributions can also change from winter to summer, for example, in IBP in summer most of the wet days are cold (Fig. 3e), but in winter there is a weak shift of the wet days towards warmer days (Fig. 3b).

## 4 The $P(T,\mathrm{WS})$ relationship

### 4.1 The dominating weather systems on wet days

Up to this point we have analysed whether wet days are predominantly warm or cold, relative to the local $T$-climatology. In this section, our objective is to identify the weather systems that occur on wet days, and, more specifically, on colder vs. warmer wet days. To this end, Figs. 3 and 4 depict, for the same regions as discussed in the previous subsection, distributions





of the dominant weather systems on wet days as a function of temperature in winter and summer, as well as the weather system relevance measure RWS introduced in Sect. 2.2. We recall that the weather systems considered are extratropical cyclones, warm

conveyor belts, atmospheric rivers, and anticyclones (potentially with convection), and that with our approach we identify for each wet day a single "dominant weather system". For details of this approach, see Sect. 2.2.

We first consider the three European regions (Fig. 3). In winter over C-EU and IBP, where wet days are predominantly warm (Fig. 2a), warm conveyor belts are most frequent on wet days, followed by cyclones and atmospheric rivers (Fig. 3a,b). Only on very few wet days (2.6%) none of the considered weather system occurred in the region, and they were attributed to

the category "NO". Cyclones are dominating on colder wet days, warm conveyor belts on moderately warm, and atmospheric rivers on the warmest wet days. Hence, the weather systems strongly influence the $P(T)$ relationship and this is reflected in RWS values of 78.3 for C-EU, which is the highest for all regions, and 68.1 for IBP (Fig. 3a,b). In contrast, in MED, cyclones dominate on most wet days independently of $T$ (Fig. 3c), and the RWS value is 44.4, indicating a smaller influence of weather systems on the $P(T)$ relationship compared to C-EU and IBP. In MED, wet days are typically cold (80%, Fig. 3c and Fig. 2a).

Warm conveyor belts contribute to some wet days in the intermediate $T$-range, whereas atmospheric rivers in a few cases dominate on warm wet days. Anticyclones are very rarely associated with wet days in all three regions.

In summer, when wet days are mostly cold in all three regions (Figs. 2b and 3d-f), the frequency and $T$-distribution of weather systems change in all regions. Over C-EU, atmospheric rivers dominate (Fig. 3d), over IBP cyclones (Fig. 3e), and over MED no weather system is clearly dominating (Fig. 3f). Note that the wet day threshold in MED in summer is very

low (Fig. 2d) such that some of the considered wet days are not associated with large amounts of precipitation. Compared to winter, a relatively large fraction of wet days cannot be assigned to a weather system, in particular in MED and IBP (labelled with "NO" in Fig. 3). We assume that precipitation on these days is related to mesoscale convective processes that frequently occur in summer in these regions, but are not attributed to our weather systems if they are not located within a well-defined anticyclone. Generally, the influence of weather systems on the $P(T)$ relationship in these regions is weaker in summer than

in winter, which is reflected in lower RWS values (Fig. 3d-f).

Considering now the three regions in North America (Fig. 4), the distribution of weather systems on wet days in winter is similar for W-US as for C-EU and IBP, but with more atmospheric rivers and fewer cyclones (Fig. 4b). In these regions, wet days are warm in winter (Fig. 2a). Over E-US, where wet days are also typically warm (Fig. 2a), atmospheric rivers are also very frequent on wet days, however, warm conveyor belts dominate on colder and moderately warm wet days (Fig. 4a). Over

C-US, the contribution of weather systems is rather complex. We recall that wet days are typically cold in this region (Figs. 2a and 4c). Anticyclones dominate on colder wet days and warm conveyor belts and cyclones on warmer wet days (Fig. 4c). However, for some wet days with moderately warm temperatures no weather system could be assigned to precipitation, which implies that other processes, for instance orographic effects, can lead to intense precipitation. Overall, in all regions, weather system frequencies depend on $T$ and are therefore important for the $P(T)$ relationship with RWS values between 60 and 65

(Fig. 4a-c).

In summer, when, as over Europe, most wet days are cold (Fig. 2b), E-US and W-US are largely dominated by atmospheric rivers (Fig. 4d,e), while anticyclonic situations mainly prevail in C-US (Fig. 4f). For both E-US and C-US, RWS values are





smaller than in winter, while for W-US, the RWS value is slightly larger. Hence, the relevance of weather systems on the $P(T)$
relationship increases for W-US, while it decreases for E-US and C-US. Interestingly, fewer summer wet days could not be
attributed to one of the four weather systems in North America than in Europe (compare Figs. 3d-f and 4d-f).

The general pattern that appears in winter in C-EU and IBP is that warm conveyor belts and atmospheric rivers dominate
more on warmer wet days whereas cyclones more on colder wet days. This appears intuitive because warm conveyor belts
and atmospheric rivers occur preferentially in the warm sector of the cyclone. Hence, precipitation on colder wet days is then
likely related to shallow convection in the cold sector of a cyclone and/or precipitation from the cloud head near the cyclone
centre. An interesting result is that anticyclones are frequently associated with intense precipitation in C-US in both seasons
(Fig. 4c,f). Plausible reasons for this result are (i) flow interactions of the anticyclones with the Rocky mountains (e.g., a
persistent, moist and upslope flow), (ii) locally embedded thunderstorms (in particular in summer), and (iii) large-scale ascent
that is often observed along the western edge of anticyclones (e.g., Zschenderlein et al., 2020). In order to better understand
the role of the involved weather systems, their potential interaction, and the general large-scale flow conditions, we now look
at the synoptic environments on the wet days.

### 4.2   Composites of weather systems on wet days

Figures 5 and 6 show composites for days when the wet day threshold (95th percentile of daily precipitation) was exceeded at
least at 50% of the grid points in the considered region. From the regions analysed in Sect. 4.1, we selected two regions each
in Europe and North America such that they represent the regional weather system variability revealed in Figs. 3 and 4. The
composites for the other regions are shown in the Supplement (Figs. S2, S3).

#### 4.2.1   Winter

Winter wet days in C-EU are associated with intense zonal flow conditions throughout the troposphere with an increased
frequency (compared to climatology) of cyclones north and anticyclones south of the precipitating area (Fig. 5a). The large-
scale weather pattern resembles the positive phase of the North Atlantic Oscillation (Hurrell et al. 2003). Recall that wet days
in this region occur on climatologically warmer days (Fig. 3a), most likely due to advection of mild (and humid) air from the
North Atlantic. Atmospheric rivers frequently occur southwest of the precipitation area, while enhanced warm conveyor belt
frequencies are found centred around the precipitation area. This strong co-location explains the dominance of warm conveyor
belts in Fig. 3a. The shift in the frequency maxima of atmospheric rivers and warm conveyor belts is in line with the classical
situation of warm conveyor belts rising from very humid low-tropospheric regions at the end of atmospheric rivers (see, e.g.,
the schematic Fig. 2.8 in Sodemann et al. 2020). Similar composite results are found for IBP and W-US (Fig. S2), with the
main differences that in these regions the potential vorticity (PV) trough is more pronounced and the positive cyclone anomaly
is more upstream. These regions also had similar $P(T,\text{WS})$ diagrams (compare Figs. 3a, 3b, 4b).

In contrast to C-EU, winter wet days in MED (which occur on colder days, Fig. 3c) are associated with a clear cyclonic wind
anomaly in the lower troposphere and cyclones occur frequently very close to the precipitation area (Fig. 5b). This agrees with
the high frequency of dominating cyclones in Fig. 3c and with the fact that the Mediterranean is a hotspot for the co-location





of cyclones and intense precipitation events (Pfahl and Wernli, 2012). The influence of warm conveyor belts is considerably reduced, and atmospheric rivers are negligible, in agreement with the low climatological frequency of these features in the Mediterranean.

As already seen in Fig. 4c, wet days in C-US are influenced by anticyclonic conditions and they also occur on colder days. Indeed, Fig. 5c reveals an increased anticyclone frequency anomaly northeast and an increased cyclone frequency anomaly southwest of C-US, which favours an anomalous lower-tropospheric easterly wind directed towards the Rocky Mountains. Thus, orographic effects influence the formation of intense precipitation in this area. The upper-level PV trough supports the formation of cyclones in the lee of the Mexican Sierra Nevada. Interestingly, CAPE values are anomalously high over the Gulf of Mexico and the anomalous southeasterly wind is directed from higher to lower CAPE values and, therefore, the orographic ascent can be associated with embedded convection. The southeasterly wind on wet days was already described by Hobbs et al. (1996) and interpreted as a low-level jet transporting moist and warm air northward. Recall that wet days are typically cold in this area, hence, the warm air advected with the low-level jet is presumably less warm than during westerly wind conditions when foehn effects lead to even warmer temperatures and dry conditions. It is also noteworthy that the 850-hPa flow direction in the C-US box and the direction of the upper-level flow, indicated by the PV-contours, are almost opposite, indicating a strong thermal wind and hence a strong meridional temperature contrast across the region.

In E-US, wet days occur ahead of an upper-level PV trough (Fig. 5d), which leads to strong southerlies transporting moist and warm air northwards from the Gulf of Mexico. Ahead of the PV trough, cyclones and warm conveyor belts evolve. Due to the high moisture transport, atmospheric rivers extend from the Gulf of Mexico across the southeastern and eastern US. Positive frequency anomalies of cyclones, warm conveyor belts and atmospheric rivers are almost collocated. The synoptic situation resembles the synoptic flash flood type in Maddox et al. (1979) and agrees with extreme precipitation events in the southeastern US driven by a pronounced upper-level trough and a steady southerly corridor of strong water vapour transport (Moore et al., 2015). CAPE values are strongly enhanced over the Gulf of Mexico and still slightly in the E-US box.

### 4.2.2 Summer

Figure 6 presents the composites for summer and reveals interesting differences to winter. In C-EU, intense precipitation occurs ahead of an upper-level PV trough and under weak cyclonic lower-tropospheric wind anomalies (Fig. 6a). Cyclones are now closer to the precipitating area, which explains the higher numbers of cyclones compared to winter (Fig. 3a,d). Another obvious difference to winter are the anomalously high CAPE values, which are generally increased over all regions considered. The composite for W-US is qualitatively similar to the one for C-EU, but with more atmospheric rivers and negligible CAPE values (Fig. S3b). The composites for the two Mediterranean regions IBP and MED reveal the influence of an upper-level trough and weak cyclonic wind anomalies and, in particular for IBP, anomalously high CAPE values (Figs. 6b and S3a). Note that over IBP cyclone frequencies are only slightly increased compared to climatology (by less than 10%) and therefore not visible in Fig. 6b.

In C-US, in sharp contrast to winter, large-scale forcing is weak in summer and convective and orographic processes seem to dominate (Fig. 6c), in agreement with Fig. 4f. The weak upper-level ridge induces an anticyclonic circulation directed towards





the Rocky Mountains. Over E-US, upper-level forcing is also weaker compared to winter, which is reflected in weaker lower-tropospheric wind anomalies (Fig. 6d). The weaker upper-level forcing and higher CAPE are typical for this season and region (Moore et al., 2015). Interestingly, E-US is located between a dipole of anomalously low CAPE values north and high CAPE values south of the precipitation area. The weak southwesterlies transport warm and humid air towards E-US, which is reflected in enhanced atmospheric river frequencies directed from the Gulf of Mexico towards the target area. Note that in this region in

summer, wet days are typically cold (Fig. 4d) and therefore warm air advected from the Gulf of Mexico is still less warm than air that is either transported from the inner continent or descending in an anticyclone, which leads to warm, but dry conditions.

In summary, considering both winter and summer, the wet day composites reveal for most regions a variant of the flow situation with an upper-level trough, a westerly or southwesterly moist flow often identified as an atmospheric river, and a surface cyclone either above the precipitation region or further northwest or north, with a warm conveyor belt directly above

the precipitating region. Interestingly, this archetypal flow configuration occurs mainly in regions and seasons where intense precipitation occurs on warmer days, but also in winter in MED and in summer in C-EU and E-US where intense precipitation occurs on colder days. This indicates that the weather system constellation is not a direct indication for whether wet days are primarily cold or warm. Very different composites were found for C-US where the flow interaction with topography appears to be essential for the occurrence of wet days in both seasons.

## 4.3   Global analysis of $P(T,\text{WS})$

Up to this point we have quantified how weather systems influence the $P(T)$ relationship in selected regions. In order to increase the validity of our findings, we repeat the analysis leading to the results shown in Figs. 3 and 4 at all grid points between 30°N and 60°N (Fig. 7). Because in summer, the 95th percentile of total precipitation can be very low, we excluded wet days with precipitation below 1 mm in Fig. 7b. In winter, wet days with $T < 0$°C are dominated by co-located cyclones, with $T$

between 0°C and 10°C by warm conveyor belts and with $T$ above 10°C, atmospheric rivers are predominantly identified as the dominant weather system (Fig. 7a). The relative fraction of anticyclones and wet days not assigned to a weather system is higher for very cold wet days below -10°C (Fig. 7a). In summer, atmospheric rivers occur more frequently in the entire $T$-distribution (Fig. 7b), hence, the results of the regional analysis (Figs. 3 and 4) are qualitatively similar to the mid-latitude analysis, except for C-US (Fig. 4c), where anticyclones occur more frequently.

Figure 7a revealed that atmospheric rivers occur frequently on warmer wet days during winter. It therefore makes sense that atmospheric rivers are often co-located with wet days in climatologically warmer areas, e.g. in the southern part of the storm tracks of the North Atlantic and North Pacific (Fig. 7c). Warm conveyor belts dominate wet days in Europe and most parts of western North America, while cyclones occur frequently in the Mediterranean and in climatologically colder regions of Eurasia and North America (Fig. 7c). Anticyclones occur frequently on wet days in climatologically dry areas near mountains, e.g. the

Rocky Mountains and the Himalayas (Fig. 7c). In summer, the influence of atmospheric rivers on wet days increases in many areas of the extratropics (Fig. 7b,d), which is in qualitative agreement with the regional analyses (Figs. 3d-f and 4d-f) and the climatologically higher frequency of atmospheric rivers in the extratropics (Sodemann et al., 2020). Areas where cyclones and warm conveyor belts are dominating on wet days are smaller compared to winter. In general, the contribution of anticyclones





to intense precipitation, potentially with convection, as described for C-US (Fig. 4f and 6c), is climatologically quite seldom in
summer (Fig. 7d). In the Mediterranean, intense precipitation can mostly not be assigned to a weather system (Fig. 7d), which
indicates that intense precipitation in this region occurs in situations with, e.g., weak pressure gradients and lower-tropospheric
instability that can lead to localised showers and thunderstorms.

We have seen in Figs. 3 and 4 that the influence of weather systems on the $P(T)$ relationship varies between the regions
and seasons, therefore, we repeated this analysis for all grid points of the extratropical northern hemisphere (Fig. 8). It can
be deduced from the regional analysis in Figs. 3 and 4 that RWS values above 50 are an indicator that considering weather
systems is relevant for understanding the $P(T)$ relationship, i.e. that $P(T)$ differs substantially for individual weather systems.
In winter, an RWS value of 50 is exceeded in the storm track regions of the North Pacific and North Atlantic, as well as over
large parts of the North American continent east of the Rocky Mountains and parts of central and western Europe (Fig. 8a). In
summer, when upper-level winds are weaker than in winter (Fig. 2a,b), the overall relevance of weather systems for the $P(T)$
relationship decreases, in particular in the North Pacific and North Atlantic, but also in parts of North America and Europe.
Hence, the $P(T)$ relationship is less influenced by the frequencies of weather systems in this area, which was already indicated
by the regional analysis (Figs. 3d-f, 4d-f). Only in some locations, weather systems are more relevant for the $P(T)$ relationship
in summer than in winter, e.g., in areas upwind of the Rocky Mountains or in western Russia (Fig. 8b). In summary, the
influence of weather systems on the $P(T)$ relationship appears to be particularly strong in regions with strong upper-level
winds, and therefore increased baroclinicity, especially in winter. However, the relevance of weather systems for the $P(T)$
relationship does not depend on the preference of wet days to occur on warm or cold days (compare with Fig. 2a,b) nor on the
wet day threshold (see Fig. 2c,d). In general, the variability of $P(T)$ for different weather systems, as measured by our relevance
metric RWS, is large in most regions of the extratropics, indicating that studying the $P(T,\text{WS})$ relationship is meaningful and
important. Exceptions are regions where a single weather system dominates wet days at all temperatures, which happens, e.g.,
with cyclones in the Arctic and in winter in the Mediterranean.

## 5 The $P(T,\text{WS})$ relationship in anomalously wet and dry seasons

The final part of this study considers the seasonal time scale and investigates how the frequency and precipitation efficiency of
weather systems change in anomalously wet and dry seasons. For this analysis, we focus on C-EU, where wet days are warm
in winter and cold in summer and essentially driven by cyclones, warm conveyor belts and atmospheric rivers (Fig. 3a,d), and
on C-US, where wet days are cold in both seasons and strongly influenced by surface anticyclones (Fig. 4c,f).

We first compare the temperature distribution on all days for the climatology with the anomalous seasons (e.g. Fig. 9a, left),
as well as the total number of days when a weather system is present in the region (e.g. Fig. 9b) and the weather systems'
precipitation efficiency (e.g. Fig. 9c), again for all vs. the wettest and driest seasons. The number of days with a weather
system is calculated as follows. First, we count the number of days a weather system overlaps with a region (green box in
Fig. 1b). If weather systems occur simultaneously, as for example in the schematic example in Fig. 1b, the weather systems
are treated as if they had occurred only half a day. If all four weather systems occur simultaneously, each system is counted





as if they had occurred only a quarter of a day. In order to increase the statistical robustness, we repeat this procedure for all 25 grid points in the region (black box in Fig. 1c), similar to Sect. 2.2. The resulting frequencies are shown as days per season (e.g. Fig. 9b). This approach enables us to quantify the contribution of each weather system to the total seasonal precipitation (e.g. Fig. 9d). The precipitation efficiency, i.e. the daily precipitation rate per weather system, is calculated for each weather system individually as the average precipitation on days when the weather system occurred in the region (e.g. Fig. 9c). Also this procedure is repeated for all 25 grid points.

## 5.1 C-EU in winter

In C-EU in winter, anticyclones are climatologically the most frequent weather systems, followed by warm conveyor belts, days with no weather system, cyclones and atmospheric rivers (grey bars in Fig. 9b). Note that in contrast to the previous sections, the frequency is calculated for all days and not only for wet days with accumulated precipitation above the 95th percentile. Although atmospheric rivers occur relatively seldom, they are associated with the highest precipitation intensity of about $7\,\mathrm{mm\,d^{-1}}$ (grey bars in Fig. 9c), while both days with anticyclones and days without a weather system have very little precipitation on average (Fig. 9c). The contribution of each weather system to the seasonal accumulated precipitation, i.e. the product of Fig. 9b and c, shows that warm conveyor belts are usually contributing by far the largest fraction to the seasonal precipitation in this region (Fig. 9d).

The wettest winter in C-EU occurred in 1994/1995 with around $420\,\mathrm{mm}$ accumulated precipitation, which is almost twice the climatological amount. This winter had an average temperature of $5.1\,^{\circ}\mathrm{C}$ and was therefore $2.1\,\mathrm{K}$ warmer than climatology. In Fig. 9a on the left, this can be seen as a clear shift of the daily mean temperature distribution towards warmer temperatures. The wettest winter is marked by more and warmer wet days than climatology (Fig. 9a, central and right). Warm conveyor belts and atmospheric rivers occur more frequently, while cyclones, anticyclones and days without a weather system occur less frequently (Fig. 9b). All weather systems are more efficient during the wettest winter, especially cyclones (Fig. 9c). One reason for the increased precipitation efficiency might be the shift to warmer wet days in this winter (Fig. 9a). Accumulated over the whole winter 1994/1995, warm conveyor belts and atmospheric rivers contribute most to the precipitation, and these two weather system categories are to a large part responsible for the extreme precipitation total (Fig. 9d, compare blue and grey bars). In the previous sections where we focused on the daily time scale, we found that for wet days, which are typically warmer (Figs. 2a), warm conveyor belts and atmospheric rivers are dominating (Fig. 3a), while cyclones are more relevant on less warm wet days (Fig. 3a). This relationship appears to have an impact also on the seasonal time scale, i.e. more warm conveyor belts and atmospheric rivers and less cyclones, together with generally warmer temperatures during the wettest winter. As already discussed in the previous section, cyclones and warm conveyor belts are always connected to each other. Hence, the lower number of days with cyclones in C-EU during the wettest winter implies that cyclones may take a slightly different track and therefore overlap less frequently with C-EU.

The driest winter in C-EU, which occurred in 2016/2017, registered about $90\,\mathrm{mm}$ of accumulated precipitation and therefore less than half of the climatological precipitation. It is slightly colder than normal seasons (Fig. 9a) and marked by, as expected, fewer wet days, and these few days are colder than normal (Fig. 9a). The signal in the frequency of weather systems is





quite clear, i.e. less cyclones, warm conveyor belts and atmospheric rivers and more anticyclones (Fig. 9b). Overall, most of the weather systems also lead to less daily precipitation and are therefore less efficient, except for cyclones, which are even slightly more efficient than climatologically expected (Fig. 9c). The majority of the seasonal precipitation is still attributed to warm conveyor belts, however, the amount is substantially lower than in the climatology (Fig. 9d). Interestingly, atmospheric
rivers contribute only very little to the seasonal precipitation, which is presumably related to the slightly colder temperatures in the driest season.

### 5.2 C-EU in summer

We now continue with the anomalous summers in C-EU. Climatologically, atmospheric rivers occur more often in summer compared to winter (Figs. 9b and 10b), but are less efficient (Fig. 10c), while warm conveyor belts occur less frequently
(Fig. 10b) but are nearly equally efficient (Fig. 10c). The seasonal precipitation is predominantly associated with atmospheric rivers, followed by warm conveyor belts and cyclones (Fig. 10d). The wettest summer occurred in 1987 with about 330 mm compared to 200 mm in the climatology and is marked by colder than average temperatures, more frequent wet days and also colder wet days (Fig. 10a). Both cyclones and atmospheric rivers occur more often than in normal seasons, warm conveyor belts less often, but all are more efficient during the wettest summer (Fig. 10b,c). The majority of the seasonal precipitation is
related to atmospheric rivers and cyclones (Fig. 10d). Warm conveyor belts are less relevant compared to winter because they in general occur less frequently in summer (Madonna et al., 2014). The driest summer occurred in 2018 with about 120 mm and is marked by fewer wet days and generally higher temperatures on all and on wet days (Fig. 10a). The driest season is characterised by fewer cyclones, warm conveyor belts, atmospheric rivers and interestingly also anticyclones, but with more days without a weather system. It is hypothesized that these days are associated with a flat SLP distribution that is not captured
by any of our weather systems. All weather systems are less efficient, except for warm conveyor belts (Fig. 10c). Warm conveyor belts contribute, in contrast to the wettest summer, most to the total seasonal precipitation, followed by cyclones, atmospheric rivers, days with no weather system, and anticyclones (Fig. 10d).

### 5.3 C-US in winter

We now compare the results for C-EU with C-US, where wet days are cold in both winter and summer (Fig. 2a,b). C-US in
winter is affected by more cyclones than C-EU, but with less warm conveyor belts and nearly no atmospheric rivers (Fig. 11b). However, all weather systems are less efficient than in C-EU (Fig. 11c), i.e. they produce less precipitation in this climatologically drier region (Fig. 2c). The wettest winter in C-US occurred in 1986/1987 with about 170 mm compared to 60 mm in the climatology and was colder than normal with colder wet days, but very frequent wet days (Fig. 11a). The frequency of weather systems does not substantially change compared to normal seasons and therefore the strongly increased precipitation efficiency
of all weather systems, apart from atmospheric rivers, which are negligible in this region, contribute to the wettest winter. Warm conveyor belts, anticyclones and cyclones are all important for the total seasonal precipitation (Fig. 11d). Interestingly, the category "NO" weather system is also substantially influencing the seasonal precipitation, which implies that some precipitation is related to smaller-scale processes, e.g. convection that cannot be identified with our large-scale weather systems. In





comparison to C-EU, the contrast between the wettest and the driest winter is very large – the driest winter, which occurred
in 2005/2006, measured only about 5 mm accumulated precipitation, which is also the reason why no wet day was identified
(Fig. 11a). Similar to the wettest winter, the frequencies of all weather systems are not substantially different from climatology
(Fig. 11b), however, all are very dry (Fig. 11c). Hence, whether a winter in C-US is anomalously wet or dry, appears to be
primarily determined by a change in the precipitation efficiency of weather systems and not by their frequency.

### 5.4 C-US in summer

In comparison to C-EU in summer, cyclones occur climatologically more frequently and atmospheric rivers less frequently in
C-US (Fig. 12b), and cyclones, warm conveyor belts and atmospheric rivers are less efficient and anticyclones more efficient
(Fig. 12c). In contrast to C-EU in both seasons and to C-US in winter, anticyclones play a more dominant role for the seasonal
precipitation (Fig. 12d), which is obvious for the wettest summer (Fig. 12d), which occurred in 1984 and registered about
315 mm rain compared to 155 mm in the climatology. Temperatures on all days in the wettest summer are colder than normal,
however, wet days are only marginally colder than wet days in normal seasons (Fig. 12a). Although the frequencies of all
weather systems change for the anomalous seasons (Fig. 12b), the precipitation efficiency appears to be the driving factor for
the precipitation anomaly (Fig. 12c). For instance, cyclones, warm conveyor belts and atmospheric rivers occur less frequently,
however, because the precipitation efficiency is substantially increased, they contribute more to the seasonal precipitation than
in normal seasons (Fig. 12d). Overall, the influence of anticyclones and the "NO" system category is strongest for the total
seasonal precipitation in the wettest season (Fig. 12d). The driest summer in 2011 with only about 30 mm rain is warmer than
normal and the very few wet days are slightly warmer than normal (Fig. 12a). Similar to the wettest season, the frequencies
of some weather systems change (Fig. 12b), but they are all very dry (Fig. 12c,d). Hence, the anomalous seasons in C-US in
summer are also largely determined by an increase or decrease in the precipitation efficiency, similar to winter.

## 6 Conclusions

The co-variability of daily and seasonal-mean surface temperature and precipitation has received much attention in the recent
years but the role of weather systems in this relationship has not been investigated yet. We therefore first quantified whether
wet days (days with precipitation above the 95th percentile) are predominantly warm or cold and then examined the relative
frequencies of weather systems on these wet days as a function of temperature. We applied this methodology to a 39-year
ERA5 climatology and to selected regions for studying processes during the wettest and driest seasons, respectively. We briefly
summarise the results of our study by addressing our research questions.

1. Does intense precipitation typically occur on the colder or on the warmer days of a season?

   Intense precipitation in winter occurs predominantly on warmer days, especially at high latitudes, whereas in the Mediter-
   ranean area and central US, intense precipitation falls predominantly on colder days. In summer, most intense rain falls
   on colder days, especially in areas close to dry summer climates, e.g. in the latitude band 35-45°N over Eurasia, but





520        also in humid summer climates like the eastern US. Only the high latitudes of the North Pacific and North Atlantic are
           dominated by intense precipitation on warmer days also in summer.

   2. Which weather systems occur preferentially on days with intense precipitation, and do some weather systems typically
      occur on the colder or the warmer wet days?

      In winter, cyclones show a slight dominance on colder wet days and are followed by warm conveyor belts on moderately
warm and atmospheric rivers on the warmest days in most areas. Central US, a region that is located near the Rocky
      Mountains, is dominated by anticyclones on colder wet days, followed by warm conveyor belts and cyclones. In summer,
      the influence of atmospheric rivers increases for most regions, in particular for western and eastern US, while for central
      US, anticyclones dominate on all wet days. The dry climates over the Iberian Peninsula and the Mediterranean have a
      comparably high number of days without any weather system, which implies that smaller-scale processes like convection
are also relevant in this area. In most of the selected regions in Europe and North America, and in particular over the
      storm track regions in the North Atlantic and North Pacific, weather systems have a stronger influence on the $P(T)$
      relationship in winter than in summer.

   3. In anomalously wet and dry seasons, how do the frequency of occurrence and the precipitation efficiency of weather
      systems differ from climatology?

The wettest winter in Central Europe is characterised by warmer than average temperatures, generally higher precipita-
      tion efficiencies of weather systems, and a higher frequency of warm conveyor belts and atmospheric rivers. During the
      driest winter, anticyclones occur more often and most weather systems are less efficient, hence, both the frequency of
      occurrence and the precipitation efficiency differ from climatology in the anomalous seasons in Central Europe. In con-
      trast, anomalous seasons in central US are characterised by a change in the precipitation efficiency and not by an altered
frequency of occurrence of weather systems. In addition, wettest seasons in central US are colder than climatology.

Our methodology involves some subjective choices and therefore there are important caveats in our study. Firstly, we rec-
ognize that our approach is sensitive to the choice of weather systems and results could therefore change when adding more
systems. However, the number of days without any weather system is quite low in most regions, at least for wet days. Only in
some regions, the fraction of days with no weather system is slightly higher, in particular mountainous regions, but still smaller
than the sum of all weather systems. In these regions, intense precipitation is furthermore associated with orographic and/or
embedded convection. Secondly, we neglected the influence of smaller-scale processes on the relationship between temperature
and precipitation. These processes could be important in regions where precipitation is primarily controlled by convection, in
particular in summer. Here it would be interesting to analyse the relation between sub-daily intense precipitation and surface
temperature in connection with weather systems, because the time scale has a huge impact on the $P(T)$ relationship itself. For
instance, sub-daily precipitation intensity can increase with surface temperature although the intensity of daily precipitation
decreases (Hardwick Jones et al., 2010; Utsumi et al., 2011).



Our results have important implications for the $P(T)$ relationship in a future climate because this relationship is, as shown in this study, influenced by the location and intensity of the upper-level jetstream in the midlatitudes and by the climatological sea ice extent in the Arctic. Both the position and intensity of the jetstream (Woollings and Blackburn, 2012), as well as the climatological sea ice extent (Johannessen et al., 2004) are expected to change with global warming. It could be interesting to examine how these two features modulate the preferred temperatures of wet days and the frequency of occurrence and precipitation efficiency of large-scale weather systems in a future climate. It is expected that in a warmer atmosphere, specific humidity and moisture transport will increase (Lavers et al., 2015), which will increase the precipitation associated with atmospheric rivers (Gershunov et al., 2019). However, we also found increased precipitation efficiencies in seasons that are colder than average, hence, a higher temperature not automatically implies higher precipitation efficiencies. Our results about $P(T,\mathrm{WS})$ in the extreme seasons clearly indicate that it is highly relevant to further study the processes that determine the precipitation efficiency of weather systems, in addition to the Clausius-Clapeyron relationship. In addition, it is not clear if and how the patterns in Fig. 2a,b change in a future climate. In general, potential changes of weather systems due to global warming and the effects of changing weather system frequency and precipitation efficiency on the $P(T)$ relationship has not yet received much attention (Barlow et al., 2019) and therefore provides avenues for further research.

*Data availability.* ERA5 data can be downloaded from https://cds.climate.copernicus.eu (last access: 23 September 2021). Weather system features are available from the authors upon request.

*Author contributions.* Both authors designed this study and discussed the results; PZ performed all analyses and wrote the paper with feedback from HW.

*Competing interests.* HW is executive editor of WCD and PZ has no competing interest.

*Acknowledgements.* The authors acknowledge funding of the INTEXseas project from the European Research Council (ERC) under the European Union's Horizon 2020 research and innovation programme (grant agreement no. 787652). We thank Michael Sprenger for providing access to the cyclone, anticyclone and warm conveyor belt features, and the INTEXseas colleagues for discussions and input, in particular Katharina Hartmuth and Mauro Hermann.





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





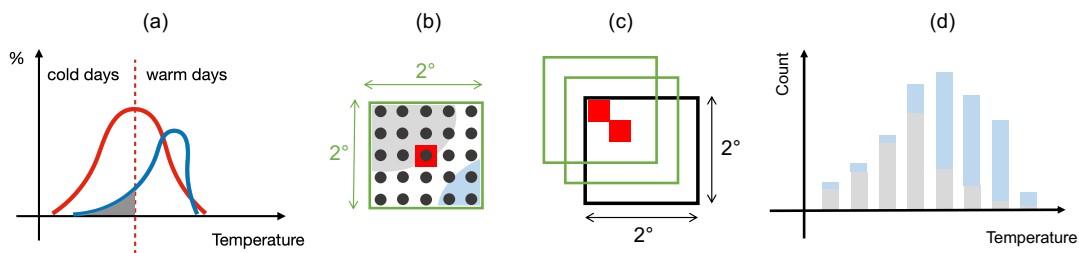

**Figure 1.** Schematics of the methodological aspects. (a) Probability distributions of daily mean 2-m temperature for all days (red) and wet days only (blue) at a particular grid point. The vertical red dashed line denotes the median of the $T$-distribution for all days, and the grey shading marks the fraction of wet days below this median. (b) A $2° \times 2°$ box is used to attribute weather systems (grey and blue shading) to precipitation at a grid point (red square). The black points symbolise each grid point in the box. (c) Analysis of (b) is repeated for all 25 grid points (black points in (b)) in the black $2° \times 2°$ box to increase statistical robustness. (d) Frequency of occurrence of the blue and grey weather systems as a function of wet-day temperature.





**Figure 2.** Fraction of wet days below the median of the local 2-m temperature distribution (WBM measure, see Section 2.1) in (a) winter (DJF) and (b) summer (JJA). Yellow crosses indicate the locations of the regions analysed later in this study, black contours show windspeed at 300 hPa starting from $25\,\mathrm{m\,s}^{-1}$ in $5\,\mathrm{m\,s}^{-1}$ increments, and the green contour marks the climatological sea ice concentration of 50 %. (c,d) illustrate the seasonal 95th percentile of daily accumulated precipitation for DJF and JJA, respectively.

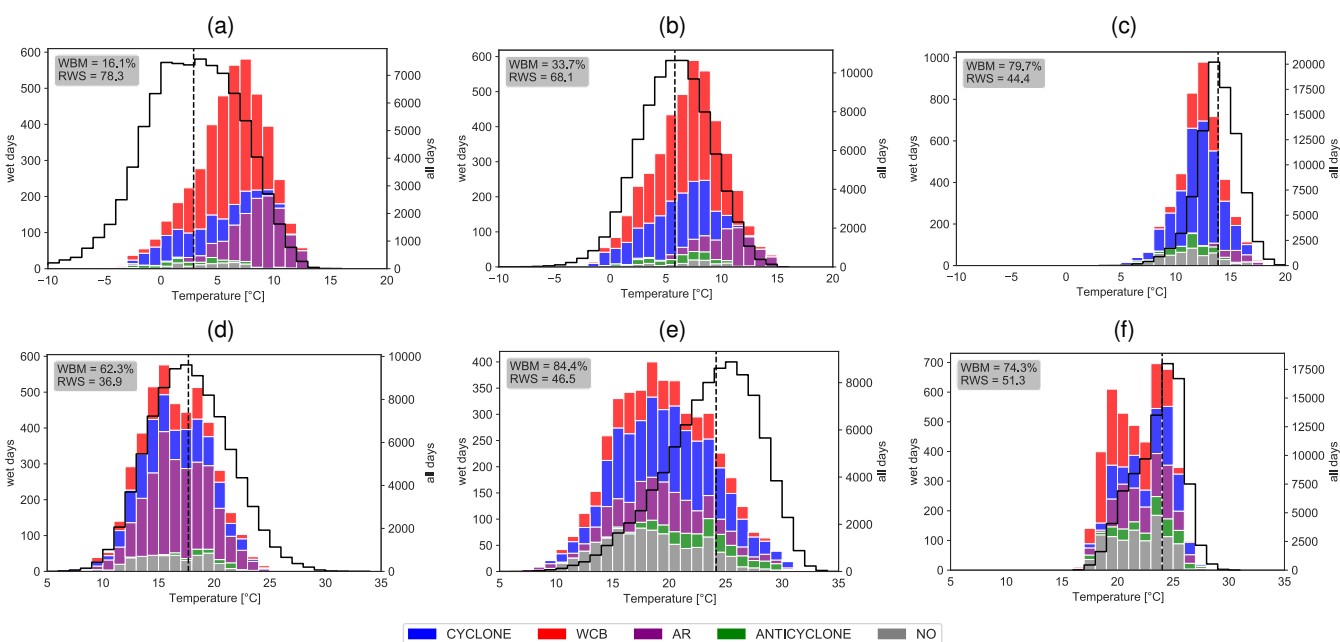

**Figure 3.** $P(T,WS)$ relationships in winter (top row) and summer (bottom row) for the following regions: (a,d): C-EU, (b,e): IBP, and (c,f): MED. The black curve shows the $T$-distribution of all days (right y-axis) and the colours in the stacked bar plots denote the dominant weather system on wet days (left y-axis). Values denote counts of grid points in the box where a particular weather system is identified as the dominant one. Note the different scaling of the y-axis in the different panels. In the upper left of each panel, values are given for the fraction of wet days below the median of daily mean temperatures (WBM, cf. Fig. 2a,b) and the relevance of weather systems for the $P(T)$ relationship (RWS).





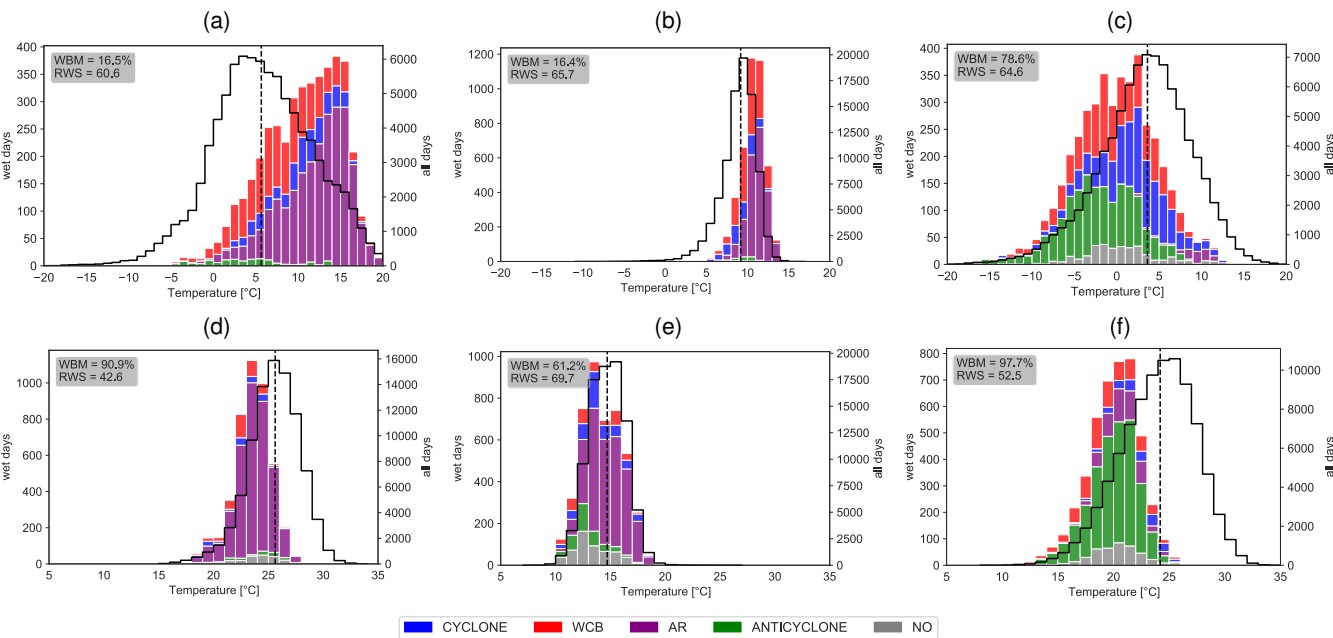

**Figure 4.** Same as Fig. 3, but for three regions in North America: (a,d) E-US, (b,e) W-US, and (c,f): C-US.

**Figure 5.** Composites for wet days in winter in the regions (a) C-EU, (b) MED, (c) C-US, and (d) E-US. Coloured contours show frequency anomalies of cyclones (blue), warm conveyor belts (red), atmospheric rivers (purple) and anticyclones (green) for values of -40, -30, -20, -10, 10, 20, 30 and 40% (dashed lines for negative values). The arrows mark the anomalous wind at 850 hPa. The black line illustrates upper-level PV at 320 K, starting from 2 pvu in 1 pvu increments. Colour shading denotes anomalies of daily maximum CAPE and the hatching marks altitudes of 1500 m. All anomalies are calculated with respect to the seasonal mean climatology. The contours of the weather features have been smoothed with a Gaussian filter.

**Figure 6.** Same as Fig. 5, but for wet days in summer in the regions (a) C-EU, (b) IBP, (c) C-US, and (d) E-US. The black contours denote PV at 340 K.

**Figure 7.** The colours in the stacked bar plots denote the dominant weather system on wet days and for all grid points from 30°-60°N in DJF (a) and JJA (b). The lower row shows the dominant weather system of all wet days in DJF (c) and JJA (d).





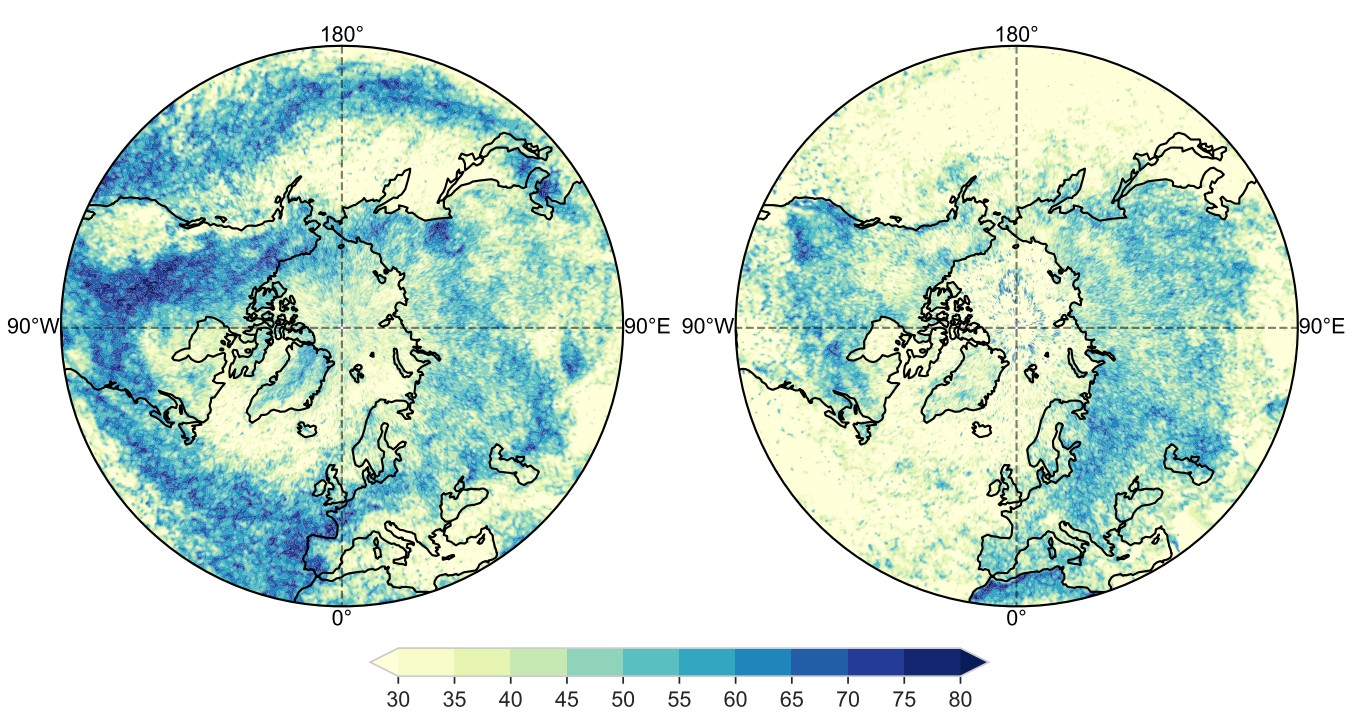

**Figure 8.** Relevance of weather systems (RWS) for the $P(T)$ relationship in DJF (a) and JJA (b).

**Figure 9.** Seasonal analysis for C-EU in winter. (a) $T$-distributions for all days and wet days only, and number of wet days, (b) mean weather system frequency on all days, (c) mean weather system precipitation efficiency, and (d) total precipitation associated with each weather system category. The colours denote results for the climatology, i.e all days (grey), and for the wettest (blue) and the driest winter (red). In (a), the orange line denotes the median, the box values are between the 25th and 75th percentile, and the whiskers extend from the 5th to the 95th percentile. Black error bars in (b)-(d) denote the inter-quartile range of the 25 grid points in the box.
**Figure 10.** Same as Fig. 9, but for summer in C-EU.

**Figure 11.** Same as Fig. 9, but for winter in C-US.





**Figure 12.** Same as Fig. 9, but for summer in C-US.




**Table 1.** Names, abbreviations and locations of the regions.

| Region | Abbreviation | Longitudes | Latitudes |
|---|---|---|---|
| Central Europe | C-EU | 4°-6°E | 48°-50°N |
| Mediterranean Sea | MED | 4°-6°E | 37°-39°N |
| Eastern US | E-US | 86°-88°W | 34°-36°N |
| Central US | C-US | 103°-105°W | 34°-36°N |
| Western US | W-US | 124°-126°W | 44°-46°N |
| Iberian Peninsula | IBP | 3°-5°W | 39°-41°N |