# Peer review of "How intense daily precipitation depends on temperature and the occurrence of specific weather systems – an investigation with ERA5 reanalyses in the extratropical Northern Hemisphere"

_Weather and Climate Dynamics, 2021_

## Author Comment (AC1)

**How intense daily precipitation depends on temperature and the occurrence of specific weather systems – an investigation with ERA5 reanalysis in the extratropical Northern Hemisphere**

Response to the Reviewers' comments by Philipp Zschenderlein and Heini Wernli

We thank both reviewers for their constructive feedback, which helped to improve the quality of the manuscript. In this document, we reply to all points raised by the reviewers. Our replies are written in blue and the comments of the reviewers in black colour.

**Reviewer 1**

I very much enjoyed reading this manuscript on the relationships between temperature, precipitation, and weather systems. Rather than examining the precipitation distributions for different temperatures, the study examines the temperature distribution for intense precipitation days. It then attributes the precipitation to the weather system that is present at the same location to determine the relevance of the weather systems to the precipitation and temperature relationship. One result that I find particularly interesting, and one that clearly requires more investigating is the fact that in particular regions, the precipitation efficiency (the average rainfall per weather system day) is increased in the wettest season, despite decreased temperatures.

Many thanks for this comment and for highlighting the interesting fact that in some regions, precipitation efficiency is increased during the wettest season, although it is colder than average. We agree that more research is needed in the future to better understand this finding, which we will emphasize more in the abstract and conclusions.

The paper is well-written and the figures are clear. I only have one main suggestion and a few typographical changes.

For the composite figures (Fig. 5 and 6), the wind, PV, and CAPE contours are shown for all wet days. These will then be most representative of the most common weather system for wet days at the location. I suggest it would be worth showing the composites for the wet days for each different weather system (even if only in the supplement). Are there other flow configurations that lead to wet days? Or do all the days with different weather systems look like the composites presented?

Many thanks for this comment, which inspired us to investigate the variability of the flow configurations on wet days. We will show composites containing wet days dominated by a particular weather system. Since we use 4 weather systems in our study (i.e., cyclones, anticyclones, warm conveyor belts, and atmospheric rivers), we will show 4 composites for each region and season plus one composite for wet days with no weather system (in the supplement).

Minor points:

Line 145-146: I don't understand what this means. How is the statistical robustness increased by doing this? Could this be explained in a different way?

We reformulated this sentence. It now reads: "In order to increase the sample size, this attribution analysis …".

Line 321: This is also shown by Hawcroft et al 2012, Catto and Dowdy 2021, Messmer and Simmonds 2021, Owen et al 2021.

Many thanks for the suggestions. We added the citations.

Line 449: "less" -> "fewer".

Changed as suggested.

Line 450: Is it not the case the WCBs are always associated with cyclones, but not necessarily the other way round?

Yes, you are right. We will reformulate this sentence.

Line 456: "less" -> "fewer".

Changed as suggested.

Line 476-477: I find the structure of this sentence difficult to read. I suggest: "Warm conveyor belts contribute most to the total seasonal precipitation, in contrast to the wettest summer, followed by…"

Reformulated as suggested.

Line 517: Suggest slight rewording. -> "Intense precipitation in winter in most regions occurs predominantly on warmer days…." Then "In summer, intense rain falls mostly on colder days…".

Thank you, reformulated as suggested.

Line 560: Suggest slight rewording: _. "hence, a higher temperature does not automatically imply…"

Reformulated as suggested.

Figure 1c: I find it unclear what is happening in this panel, and the text does not help - could this be explained differently?

You are right, Fig. 1c is confusing, actually, it is unnecessary. It only illustrates that the analysis for the red grid point in Fig. 1b is repeated for all 25 grid points (i.e., the black points in Fig. 1b) of the green box. We therefore decided to remove Fig. 1c. In addition, we slightly adapted a sentence in Sect. 2.2. It now reads: "In order to increase the sample size, this attribution analysis is not only performed for the red grid point in Fig. 1b, but for all 25 grid points (black points in Fig. 1b) enclosed by the green 2°x 2° box."

General comments:

I found this manuscript to be generally well-written and clear. The graphics provide clear illustrations of the main results of the study. The topic of the study has substantial scientific merit, and I expect it will be of interest to readers of WCD. I have a number of minor comments for the authors to consider. Once these comments are satisfactorily addressed, I believe this manuscript may be acceptable for publication.

Specific comments:
L83–86: "The combined effects...precipitation." This statement needs references. Also, did you intend to say "from the southeast" instead of "from the southwest" here?
We added the review paper of Lin et al. (2001), and yes, we mean "from the southeast". Thank you for the careful reading.

L148–149: A shortcoming of this method is that it does not account for the remote influences that the weather systems can have on precipitation, i.e., the fact that a weather system, such as a cyclone, can still have a significant influence on precipitation at a given location even if that system is not identified close to the location. For instance, a cyclone, as identified using the method of Wernli and Schwierz (2006), may be located well to the west of a location but may still drive moisture transport and forcing for ascent that contributes to the precipitation at that location. Similarly, anticyclones often play a key role in driving moisture transport associated with atmospheric rivers associated with extreme precipitation, but do not necessarily overlap the region of precipitation. Perhaps you can comment on these issues in the text?
You are right that this is one shortcoming of our approach, and we agree that also remote weather systems can be important for intense precipitation events. However, for the P(T,WS) relationship analysis in Figs. 3 and 4, it is difficult to estimate remote influences. An idea would be to increase the area that is used to attribute the weather system to intense precipitation events. For example, one could increase the size of the green box in Fig. 1b. This larger area would incorporate also remote weather systems, however, it is not straightforward to estimate to what degree this remote weather system influences the local intense precipitation event. We therefore think that the composite analysis in Figs. 5-6 is sufficient to analyse remote influences on intense precipitation events. We thank the reviewer for his/her comments and we added some comments on this issue at the end of the third paragraph of Sect. 2.2.

L208–209: I find this statement confusing. I understand the argument that high baroclinicity can favor intense precipitation on warm days, but I do not understand how low baroclinicity provides a favorable condition for intense precipitation on colder days. Please clarify. Perhaps the occurrence of intense precipitation on colder days relates to a tendency for the precipitation in those locations to involve orographic forcing?
This sentence was indeed a bit unclear and misleading; we therefore deleted the second part of the sentence.

L217: Other factors that may contribute to this relationship include surface cold pool formation associated with moist convection, decreased insolation due to increased

cloudiness, adiabatic cooling due associated with orographic ascent, and the occurrence of precipitation in connection with the passage of weak cold fronts.

Thank you for these valuable suggestions, we included them in the text.

L294: Orographic forcing might also be an important process.

We did not include orographic forcing in this sentence because our intention was to only describe the relevance of warm conveyor belts, atmospheric rivers, and cyclones for warm/cold wet days, since they were the most relevant weather systems in most regions. Orographic forcing is listed as an important process in the following sentences, where we describe the importance of anticyclones for wet days in C-US.

L308: How were the climatological relative frequencies for the weather systems computed? Are they computed using all days during Dec–Feb 1980–2019? Please clarify.

Yes, we added a sentence at the end of Sect. 4.2: "Anomalies in Figs. 6, 7, S2, and S3 are calculated with respect to the seasonal mean climatology, which incorporates all days during DJF and JJA 1980-2019, respectively."

L330–331: This statement is vague. Can you be a bit more specific about what Hobbs et al. found in their study?

We included the following sentences: "Hobbs et al. (1996) found that the convergence of subsiding air off the Rockies, which is warm and dry, and the warm and moist air transported northwards from the Gulf of Mexico can lead to a narrow zone of strong moisture gradients – a dryline. If the dryline co-occurs with a lee trough, convective precipitation and thunderstorms can develop (Hobbs et al., 1996)."

L332: It is perhaps worth mentioning that this pattern is consistent with the occurrence of a cold surge along the Rocky Mountains (e.g., Colle and Mass 1995). Such cold surges involve cold air damming and the establishment of a strong sloping baroclinic zone. Precipitation within the cold air mass could be associated upslope flow along the mountains as well as ascent of warm conveyor belts along the sloping baroclinic zone. Colle, B. A., and C. F. Mass, 1995: The structure and evolution of cold surges east of the Rocky Mountains. Mon. Wea. Rev., 123, 2577–2610, https://doi.org/10.1175/1520-0493(1995)123<2577:TSAEOC>2.0.CO;2.

Thank you for the suggestion. We included the citation.

L337: This sentence is a bit confusing. What do you mean by 'evolve' here? Can you be more specific?

This sentence now reads: "Cyclones and warm conveyor belts are located ahead of the PV trough."

L359–361: I am having difficulty following this physical explanation.

We deleted the "warm" in "… therefore, warm air advected from the …".

L546: To which "smaller-scale processes" are you referring here? Be specific.

We added "cloud microphysics".

Technical corrections:
L88: "CAPE" has not been defined in the text.

We now included the definition.

L89: Remove comma after "analyse"?
Removed.

L320: Remove "dominating"
Removed.

L328: Change "Nevada" to "Madre"
Changed as suggested.

L328–330: "Interestingly ... convection." This seems to be a run-on sentence.
We split the long sentence into two sentences.

L388: Insert "those in" before "winter"
Inserted as suggested.

L496: Insert "are" after "anticyclones"
Inserted as suggested.